# Point-of-Care Ultrasound-Guided Protocol to Confirm Central Venous Catheter Placement in Pediatric Patients Undergoing Cardiothoracic Surgery: A Prospective Feasibility Study

**DOI:** 10.3390/jcm10245971

**Published:** 2021-12-20

**Authors:** Torsten Baehner, Marc Rohner, Ingo Heinze, Ehrenfried Schindler, Maria Wittmann, Nadine Strassberger-Nerschbach, Se-Chan Kim, Markus Velten

**Affiliations:** 1St. Nikolaus-Stiftshospital Andernach, Ernestus-Platz 1, 56626 Andernach, Germany; torsten.baehner@stiftshospital-andernach.de; 2Department of Anesthesiology and Intensive Care Medicine, Rheinische Friedrich-Wilhelms-University, University Hospital Bonn, Venusberg-Campus 1, 53127 Bonn, Germany; Marc.Rohner@ukbonn.de (M.R.); Ingo.Heinze@ukbonn.de (I.H.); ehrenfried.schindler@ukbonn.de (E.S.); maria.wittmann@ukbonn.de (M.W.); nadine.strassberger-nerschbach@ukbonn.de (N.S.-N.); se-chan.kim@ukbonn.de (S.-C.K.)

**Keywords:** central venous catheter 1, point-of-care ultrasound 2, pediatric cardiac anesthesia 3

## Abstract

Background: Central venous catheters (CVC) are commonly required for pediatric congenital cardiac surgeries. The current standard for verification of CVC positioning following perioperative insertion is postsurgical radiography. However, incorrect positioning may induce serious complications, including pleural and pericardial effusion, arrhythmias, valvular damage, or incorrect drug release, and point of care diagnostic may prevent these serious consequences. Furthermore, pediatric patients with congenital heart disease receive various radiological procedures. Although relatively low, radiation exposure accumulates over the lifetime, potentially reaching high carcinogenic values in pediatric patients with chronic disease, and therefore needs to be limited. We hypothesized that correct CVC positioning in pediatric patients can be performed quickly and safely by point-of-care ultrasound diagnostic. Methods: We evaluated a point-of-care ultrasound protocol, consistent with the combination of parasternal craniocaudal, parasternal transversal, suprasternal notch, and subcostal probe positions, to verify tip positioning in any of the evaluated views at initial CVC placement in pediatric patients undergoing cardiothoracic surgery for congenital heart disease. Results: Using the combination of the four views, the CVC tip could be identified and positioned in 25 of 27 examinations (92.6%). Correct positioning was confirmed via chest X-ray after the surgery in all cases. Conclusions: In pediatric cardiac patients, point-of-care ultrasound diagnostic may be effective to confirm CVC positioning following initial placement and to reduce radiation exposure.

## 1. Introduction

Central venous catheters (CVC) are commonly required in neonatology and pediatric intensive care patients, and ultrathin single-lumen catheters are preferably used. However, during high-risk procedures, such as congenital cardiac or major pediatric surgeries, multi-lumen CVC with larger diameters, analogous to those used in adult perioperative medicine, are required to perform differentiated drug therapies and perfuse higher volumes. While various central veins are feasible to access, the most common puncture site for CVC in pediatric anesthesia for congenital heart surgery in Germany is the internal jugular vein [1]. Point-of-care verification of CVC positioning is not standardized in pediatric patients and correct positioning is usually verified postoperatively by X-ray [2]. However, incorrect positioning of the catheter may induce serious complications, such as extravasation, pleural and pericardial effusion, arrhythmias, valvular damage, or incorrect drug release before correct CVC positioning is verified post-surgery [3,4]. Therefore, timely point of care diagnostics, preferably without exposing the patient to radiation, and position verification immediately after introduction or during insertion may have enormous advantages and would be desirable for correct positioning of the catheter tip, preventing potential deleterious consequences. 

Ultrasound-guided techniques of correct CVC tip positioning, e.g., a method of direct ultrasound tip position of CVC using a microconvex probe, have been described in adult perioperative patients and are already established in the clinical routine [5]. 

Central venous puncture is nowadays ultrasound guided at the majority of pediatric heart centers [1]. However, existing formulas calculating depth of insertion based on body dimensions are unreliable, especially in children with pathologic heart dimensions or abnormal anatomy [6]. For example, in cases of extreme cardiomegaly, the CVC may be inserted too deep after only a few centimeters.

At our center, puncture of the central vessel is also performed under continuous ultrasound guidance. Our research group has found that ultrasound can also be used to monitor the intrathoracic position of the CVC tip, especially in young children. Jugular, parasternal, and subcostal ultrasound windows have proven to be favorable. 

The relatively large glandular tissue of the thymus provides a very good acoustic window for the central intrathoracic vessels, especially in young children who have not undergone previous thoracic surgery. It is also favorable that the large thymic tissue in young children displaces the lung parasternal and therefore provides a good view of the central vessels. Another favorable circumstance is that the thorax in the infant is largely cartilaginous, and thus no sound cancellation by bone complicates ultrasound diagnosis. 

Therefore, the aim of the present study was to test the hypothesis that CVC placement and tip confirmation can safely be performed via point-of-care ultrasound visualization, reducing radiation exposure in pediatric patients undergoing cardiothoracic surgery.

## 2. Materials and Methods

To evaluate the safety and feasibility of ultrasound-guided CVC placement and tip positioning in pediatric patients undergoing cardiothoracic surgery, the following examination protocol has been developed based on our daily routine. The ideal position of the CVC tip is the junction of the superior vena cava (SVC) and the right atrium (RA). Consecutively, from the two parasternal, the suprasternal notch, and the subcostal view, the SVC, the CVC tip itself, and the right pulmonary artery (RPA) have been attempted to be identified. If the junction of the superior vena cava into the right atrium could not be identified, the RPA was used as an accessory structure. The RPA is in direct anatomic relation to the junction of the superior vena cava with the right atrium [7].

The present prospective study was conducted at the pediatric cardiac surgery suite and pediatric cardiac ICU at the University Hospital Bonn, Germany. The study was performed in accordance with the principles expressed in the Declaration of Helsinki and after approval by the institutional revenue board (IRB) at the Rheinische Friedrich-Wilhelms-University Bonn (protocol No. 159/14). Nineteen children, ranging from 3 days old to 4 years of age and weighing between 2.55 and 16 kilograms in body weight, that underwent pediatric cardiac surgery for congenital heart disease, were included. The sample size was calculated based on previous studies’ evaluations on CVC tip positioning in adult patients [5].

Children requiring a CVC for cardiac surgery were included and evaluations were performed by anesthesiologists highly experienced in both pediatric cardiac anesthesia and vascular ultrasound. A total of 28 examinations were performed on 19 children. Ten children were examined by one examiner, and nine children were examined by two examiners and were assessed separately.

After induction of general anesthesia, the children received a CVC according to the standard in-hospital procedure using ultrasound-guided puncture of the right jugular venae. The primary depth of insertion of the CVC was calculated based on the child’s height using the following formula: primary insertion depth (cm) = body length (cm)/10. Subsequently, using a standard examination procedure with the above-mentioned 4 ultrasound views, an attempt was made to identify the following structures: superior vena cava, right pulmonary artery, and the CVC tip (Figure 1). 

The examiner had to rate the visualization of the identified structures and duration of the assessment on a five-point scale. This five-point scale was defined as follows:
Class 1Structures can be reliably identified, sharp and complete contour, prompt displayClass 2Structures can be delineated, not reliably complete, no prompt displayClass 3Structures incompletely identified, only after image optimization, delayed visualizationClass 4Structures insufficiently identified after image optimization, strongly delayed visualizationClass 5Structures not visualized

After distinct visualization of the CVC tip in the SVC, the catheter was corrected for position.

Echo views: here, we describe the 4 probe positions and subsequent views that were evaluated in the present manuscript for CVC placement and tip confirmation (Figure 1). 

First view: parasternal craniocaudal view

A linear transducer, or hockey stick, was used for the assessment of the parasternal view. The transducer was positioned in the 3rd intercostal space right parasternal with the transducer index mark parallel to the sternum. The superior vena cava (SVC) is present in the longitudinal section; the junction of the SVC with the right atrium is depicted and the CVC is visible in the SVC (Figure 2).

Second view: parasternal transversal view

A linear transducer, or hockey stick, was used. From the previous position, the probe was rotated 90 degrees. The SVC was visualized in the cross-section, with the brachiocephalic trunk (BCT) and the right carotid artery. The right pulmonary artery (RPA) can be seen longitudinally (Figure 3).

Third view: suprasternal notch view

A cardiac sector transducer was primarily used. The use of a linear transducer/hockey stick is also possible in young children. The limiting factor is the penetration depth of the linear transducer or hockey stick. The connection from the right internal jugular vein to the innominate vein (Innom V) can be seen. Ideally, the junction into the right atrium and the distal ascending aorta in the cross-section are also visualized (Figure 4).

Fourth: subcostal view

A cardiac sector transducer was primarily used. The use of a linear transducer/hockey stick might also be possible in young children. The limiting factor is the penetration depth of the linear transducer or hockey stick.

View from subcostal, a four-chamber view can be seen. By tilting the transducer, the superior vena cava is displayed in a longitudinal section, and the entry of the superior vena cava into the right atrium is clearly visible (Figure 5). A PW Doppler signal can be placed in the confluence of the superior vena cava to the atrium and a vena cava flow signal can be detected.

## 3. Results

The present study included children aged between 2 days and 54 months. The mean age was 8.9 months, and the mean body weight was 5.8 ± 3.5 kg with a range between 2.5 and 16 kg (Table 1). In all children, the CVC was successfully inserted into the right internal jugular vein. In all cases, the position of the CVC was verified postoperatively by chest X-ray. Additionally, positioning was intraoperatively verified by the surgeon through visual inspection during opening of the cardiac structures, or by palpation of the CVC tip in the SVC. The mean insertion depth was 0.13±0.02cm/cm body height (Table 1).

Ten children had no previous surgical procedures, while the remaining nine children had undergone surgery previously. Due to the procedure or individual reasons, not each of the four views could be obtained in all children. In one child, the parasternal and jugular views were not performed, and in two other children, the suprasternal view was not performed. However, most views have been performed in all patients included in our study. The most successful visualization of the superior vena cava was from subcostal (88%), followed by the suprasternal notch (87%), and the parasternal transversal view in (74%) of the cases (Table 2).

Although visualization of the SVC from the parasternal transversal view and suprasternal notch view was easy to achieve, the tip of the CVC could only be identified in 41% and 22% of the cases, respectively. Similarly, identification of the right pulmonary artery was possible from the suprasternal view in only 26% of the cases.

The most reliable visualization of all structures from one view was achieved from the subcostal view. From the subcostal view, the superior vena cava was visible in 88% of the cases, the CVC tip in 79%, and RPA in 75% of the cases.

Particularly in pre-operated children, the parasternal views were significantly more difficult or even impossible to achieve. The visualization of the structures in these cases was more frequently classified as poor or not possible. Thus, in none of the nine children with previous sternotomy, neither the CVC tip, nor the right pulmonary artery could be reliably visualized in the parasternal views (Figure 6a,b).

In summary, using the combination of the four views, all structures have clearly been visualized in the majority of the children and the CVC tip could be identified and positioned in 25 of 27 examinations (92.6%). Correct positioning was confirmed via chest X-ray after the surgery in all cases.

## 4. Discussion

Ultrasound-guided puncture of central vessels is an accepted standard procedure for the insertion of central venous catheters in adult as well as in pediatric patients, preventing incorrect puncture and damage to proximate structures as well as arterial puncture [8,9,10,11].

Formulas calculating the insertion depth are commonly used in pediatric patients but are very imprecise. The reasons for this may be due to the patient, e.g., cardiomegaly due to heart disease, or to the physician performing the procedure, e.g., due to different insertion points [6,12]. Therefore, verification of the correct insertion depth and positioning is still performed by X-ray diagnosis in many areas and subsequent correction of the CVC positioning is frequently required.

Ultrasound-guided positioning of the CVC tip has been described in adults, but studies in children have not been performed so far [5,13,14]. The present feasibility study utilizes four transthoracic views visualizing the tip of the CVC and the related anatomical structures. In children, from the suprasternal notch, the SVC can be visualized very easily.

However, for reliable identification, a pulsed-wave Doppler signal can be derived from the superior vena cava, especially in the case of subcostal sections. This is usually possible and prevents confusion, e.g., with the brachiocephalic trunk (Figure 7).

From the subcostal view, the distance between the CVC tip and the junction of the superior vena cava and the right atrium can usually be measured precisely. The CVC can thus be precisely retracted to the level above the right pulmonary artery. This offers significant advantages, especially in procedures where ligation of the superior vena cava is required, such as a Glenn operation. From the subcostal view, the essential structures can be identified in both naive children and children that have undergone previous cardiac surgery. If only one view is intended to be used for positioning of the CVC tip, the subcostal view is most promising in verifying CVC tip position. However, the results of our study indicate that, by combining the four views, a very good orientation and the correct intravascular position as well as the correct insertion depth of the CVC can be achieved during insertion of the central venous catheter.

Difficulties identifying the anatomic structures and CVC tip have been observed in children who have previously undergone cardiac surgery. The thymus provides an ideal parasternal acoustic window in small children. Therefore, these difficulties may be due to the fact that the thymus, and its favorable acoustic conditions, has been removed in the previous surgery and is therefore no longer present. In addition, there is certainly scar tissue which also restricts the view from parasternal views.

The CVC is identified by visualizing a double structure in the vein. The tip of the CVC is particularly echogenic. If the identification of the CVC tip is not clear, a useful tool is continuous flushing of the CVC with saline. Due to the small diameter of the CVC, a turbulent flow is created at the CVC tip, so that a clearly visible jet is created even without the use of agitated saline or ultrasound contrast medium (see Video S1). The jet is easy to identify and provides an indirect indicator of the position of the CVC. In addition, it is often possible to identify the J-wire of the CVC passing over the tip of the CVC (Figure 8).

Most importantly, in all cases when the tip positioning was feasible through echocardiographic evaluation, the correct positioning was confirmed intraoperatively by direct visualization or palpation as well as postoperatively via X-ray verification of the CVC tip at the junction of the SVC and the RA. However, the examinations in our study were performed by colleagues highly experienced in cardiac and vascular ultrasound evaluations and it is unclear how much training is required to gain competence for sonographic CVC evaluation. Therefore, further investigations of the views in a larger group of children and examiners are required, evaluating the safety and precession of ultrasound-guided CVC insertion depth in children, and verifying the results of this feasibility study. Furthermore, chest X-ray diagnostics are performed after cardiac surgery, not only evaluating the lungs and confirming CVC tip positioning, but also evaluating the heart and chest wall. However, during ICU therapy, the CVC often needs to be replaced and our protocol may have the potential to reduce radiation in situations when radiography diagnostics are only performed for tip CVC positioning verification and pneumothorax exclusion.

Medical radiation is the largest source of radiation exposure accounting for a mean effective dose (ED) of 3.0 mSv/y per person originating from diagnostic and therapeutic interventions. The annual ED in pediatric patients with CHD is relatively low (<3 mSv/y); however, yearly exposure accrues over the lifetime, potentially reaching high values (>100 mSv) in selected cohorts of chronic pediatric patients and cancer risk estimation highlights the need to limit medical radiation exposure [11,15].

## 5. Conclusions

Based on the present observational feasibility study, we were able to demonstrate that ultrasound-guided CVC insertion and tip positioning in children is a fast and reliable technique, preventing unnecessary X-ray exposure that may be considered for future policies.

## Figures and Tables

**Figure 1 jcm-10-05971-f001:**
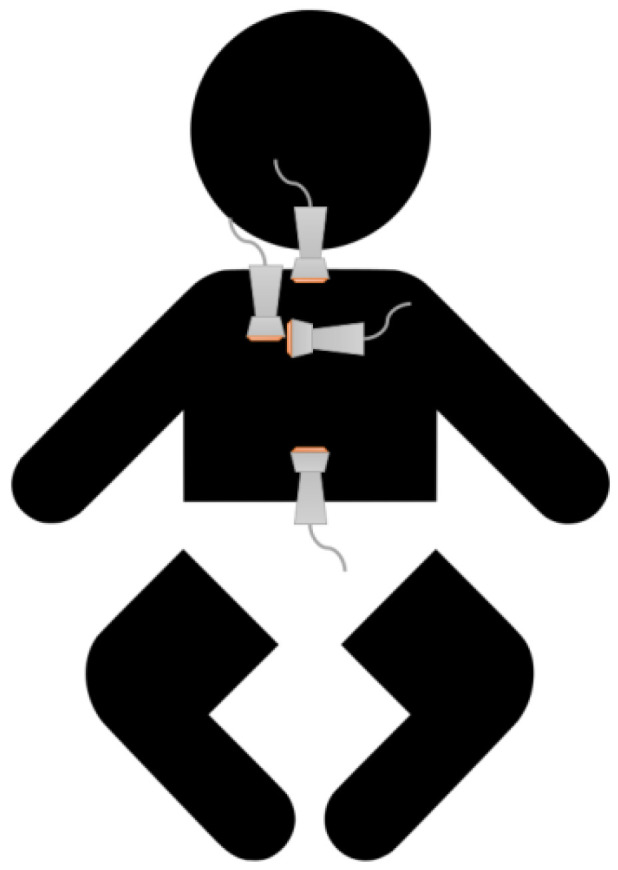
Schematic view of the four probe positions.

**Figure 2 jcm-10-05971-f002:**
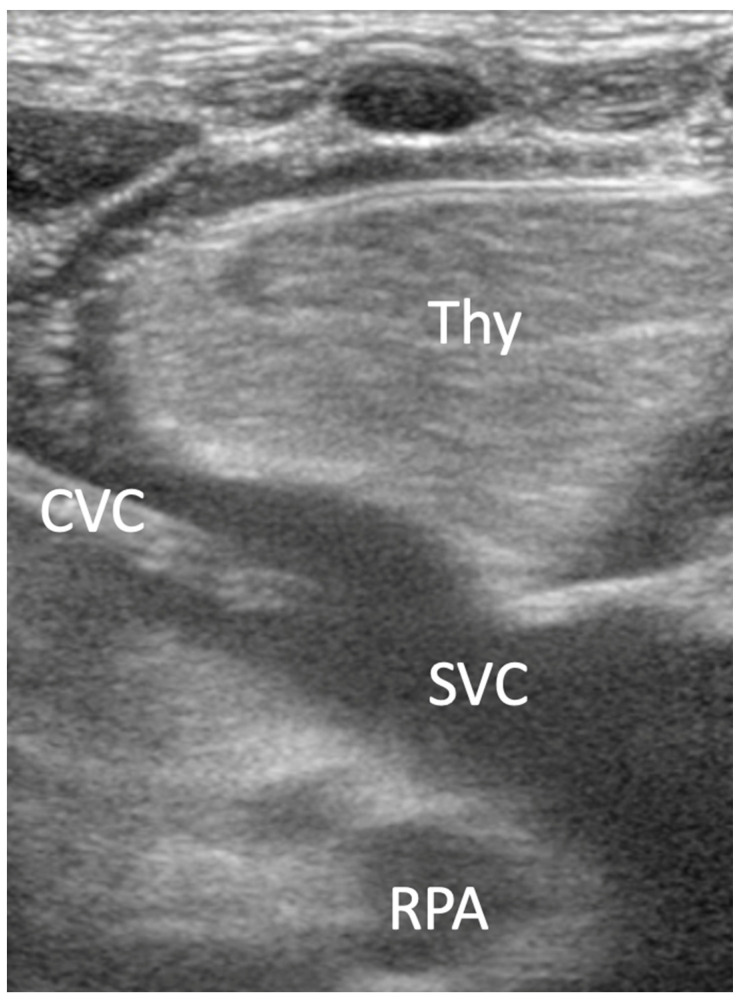
Parasternal craniocaudal view: thymus (Thy), central venous catheter (CVC), superior venae cava (SVC), right pulmonary artery (RPA).

**Figure 3 jcm-10-05971-f003:**
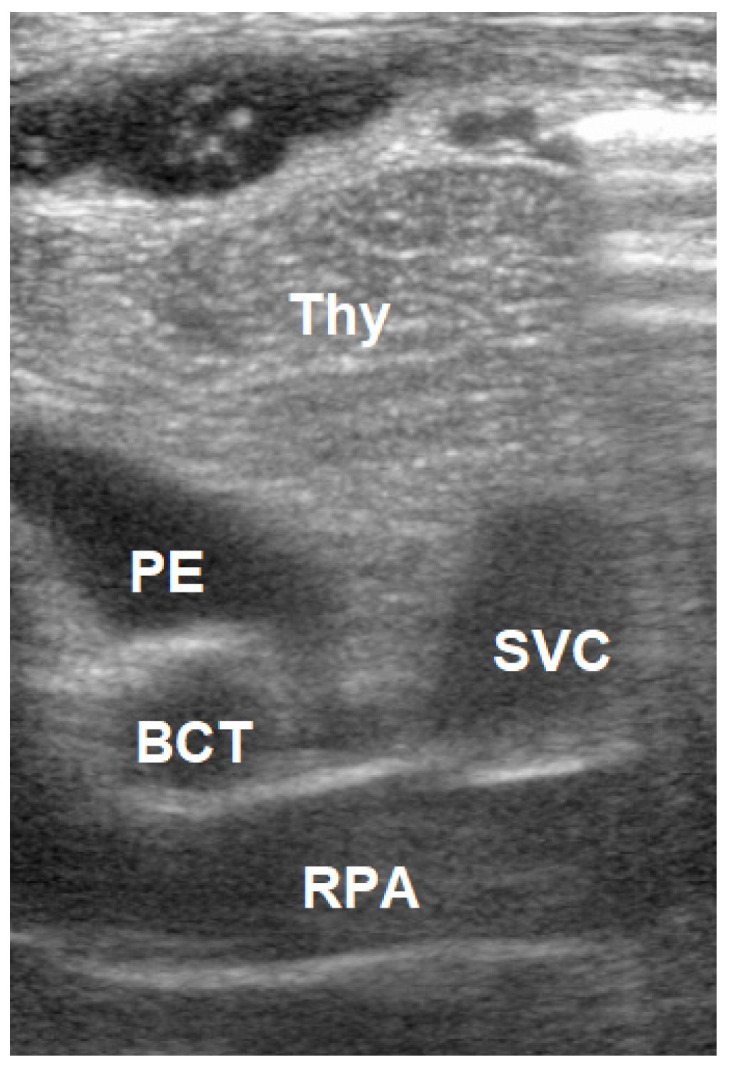
Parasternal transversal view: superior venae cava (SVC), brachiocephalic trunk (BCT), right pulmonary artery (RPA), pericardial effusion (PE).

**Figure 4 jcm-10-05971-f004:**
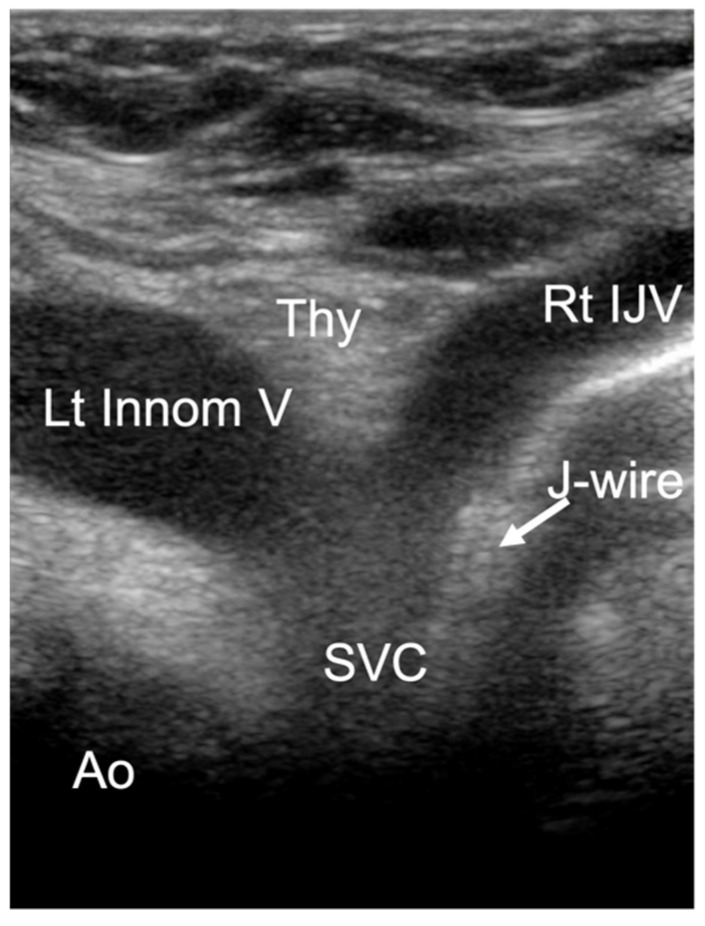
Suprasternal notch view: superior vena cava (SVC), right internal jugular vein (Rt IJV), left innominate vein (Lt Innom V), aorta (Ao), thymus (Thy).

**Figure 5 jcm-10-05971-f005:**
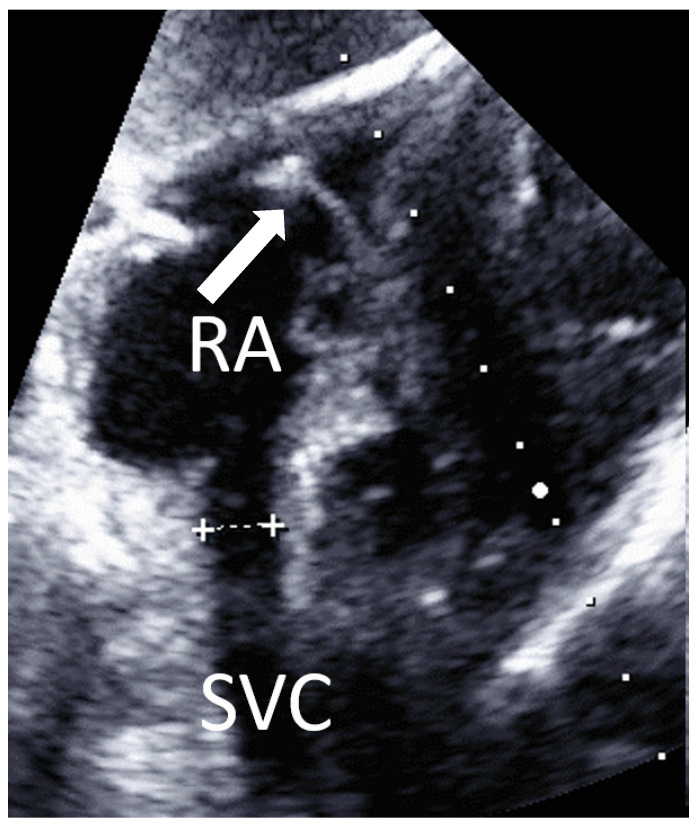
Subcostal view: superior vena cava (SVC) and right atrium (RA). Arrow: J-wire in RA. Caliper indicates SVC diameter.

**Figure 6 jcm-10-05971-f006:**
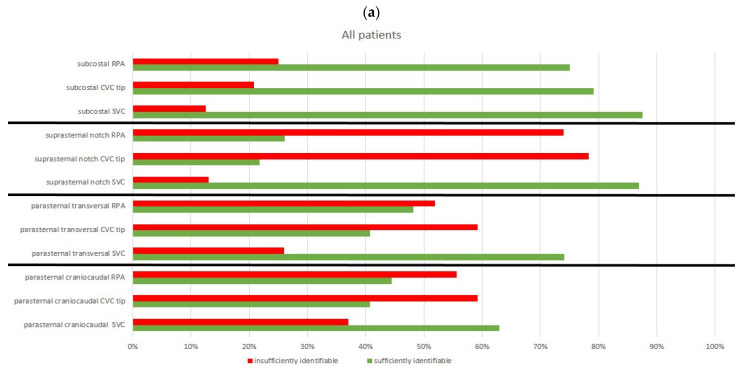
Visualization of CVC tip and anatomical structures in all patients (**a**) and patients with previous surgery (**b**), (% all patients in respective group).

**Figure 7 jcm-10-05971-f007:**
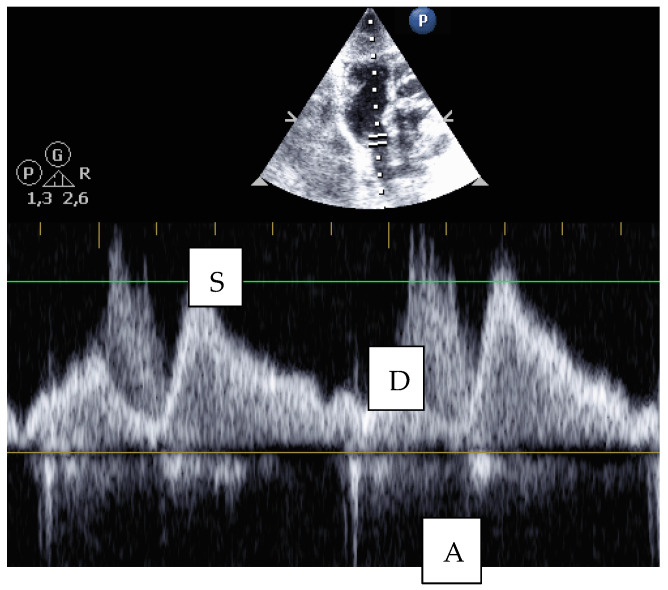
Subcostal view: PW Doppler signal from the superior vena cava with systolic (S), diastolic (D), and atrial (A) waves.

**Figure 8 jcm-10-05971-f008:**
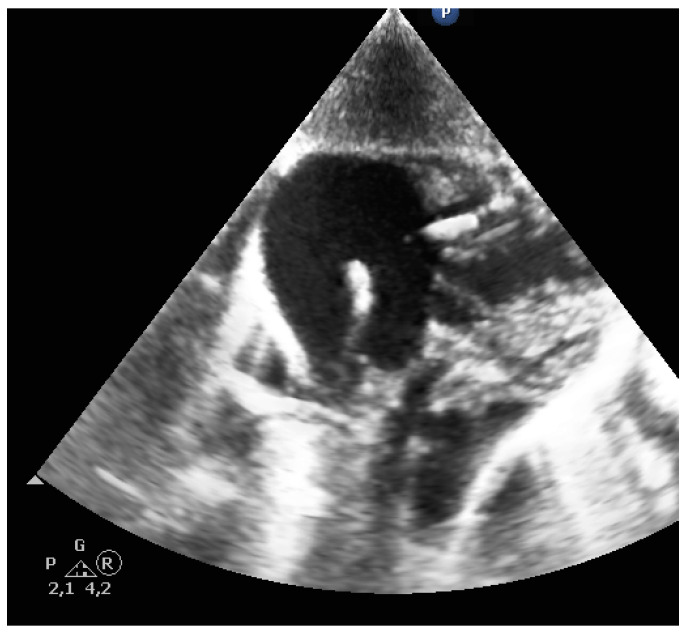
J-wire of the CVC passing over the tip of the CVC.

**Table 1 jcm-10-05971-t001:** Patients’ characteristics.

	Mean	SD	Range
body weight (kg)	5.8	3.46	2.55–16
body height (cm)	65	16	50–109
age (months)	8.9	15	0.007–53.37
insertion depth (cm)/body height (cm)	0.13	0.02	

**Table 2 jcm-10-05971-t002:** Echocardiographic evaluation.

Parasternal Craniocaudal View	*n*	Sufficiently Identifiable		Mean Classification	SD
SVC	27	17	63%	1.94	17
CVC tip	27	11	41%	1.64	11
RPA	27	12	44%	2.25	12
Parasternal transversal view	*n*	sufficiently identifiable		mean classification	SD
SVC	27	20	74%	2.10	20
CVC tip	27	11	41%	1.73	11
RPA	27	13	48%	2.31	13
Suprasternal notch view	*n*	sufficiently identifiable		mean classification	SD
SVC	23	20	87%	1.75	20
CVC tip	23	5	22%	2.60	5
RPA	23	6	26%	2.17	6
Subcostal view	*n*	sufficiently identifiable		mean classification	SD
SVC	24	21	88%	1.38	0.65
CVC tip	24	19	79%	1.53	0.68
RPA	24	18	75%	2.17	0.76

Superior vena cava (SVC), central venous catheter (CVC) tip, right pulmonary artery (RPA).

## Data Availability

The data presented in this study are available on request from the corresponding author.

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
