# Peer review of "Point-of-Care Ultrasound-Guided Protocol to Confirm Central Venous Catheter Placement in Pediatric Patients Undergoing Cardiothoracic Surgery: A Prospective Feasibility Study"

_jcm, 2021, doi:10.3390/jcm10245971_

Round 1
Reviewer 1 Report
This study of Baehner et al. is an interesting research that suggests the use of ultrasound-guided CVC placement in paediatric patients undergoing cardiothoracic surgery to reduce radiation exposure.
However, I am not sure whether this work has to be published as an “article”. I consider it would be more suitable to be published as a protocol or concept paper. I suggest authors to format their manuscript according to guidelines for this article types.
My only concern is about the use of X-ray to confirm CVC position after the surgery, as authors stated that reducing patient exposure to radiation would be beneficial and achievable using ultrasound. So, despite using ultrasound, do patients will need to undergo X-ray to confirm CVC position or it was only performed in this study to validate their results? In case ultrasound is finally used as a standard for CVC placement, would it be also needed to performed X-ray likewise?
Minor issues:
- Abstract: line 14: remove “potentially” before cardiogenic.
- Abstract: line 16: “can be performed fast and safe by…”
- Abstract: line 17: please define “IRB”.
- Abstract: line 17: “inform consent obtainment”
- Abstract: line 18: “four probe positions”
- Abstract: line 19: “cardiothoracic surgery”
- Abstract: line 23: following initial placement, and to reduce radiation exposure”
- Abbreviate central venous catheter as CVC all through the manuscript: see lines 22, 81, 197, 208, 225.
- Introduction: line 35: standardized.
- Results: line 182: “the CVC tip or the right pulmonary artery could…”
- Discussion: lines 197 and 207. are contradictory: at the beginning authors stated that ultrasound-guided puncture of central vessels is an accepted standard procedure for the insertion of CVC in adults as well as in pediatric patients (7-9). Then, they stated that ultrasound-guided positioning of the CVC tip has been described for adults, but it is not a standard procedure in children (5, 12). I guess it is not accepted as a standard procedure in children, so authors must correct it and clarify.
- Discussion: line 210: Please define “PW”.
- Figures 6a, 6b: Please detailed to which aspect percentages refer to.
Author Response
This study of Baehner et al. is an interesting research that suggests the use of ultrasound-guided CVC placement in paediatric patients undergoing cardiothoracic surgery to reduce radiation exposure.
To begin with we would like to thank the reviewers for the time he invested into our submission and for his valuable suggestions that we addressed in a point-by-point response.
However, I am not sure whether this work has to be published as an “article”. I consider it would be more suitable to be published as a protocol or concept paper. I suggest authors to format their manuscript according to guidelines for this article types.
We feel that the special issue "Congenital Heart Disease: New Insights in Diagnosis and Clinical Management" is the perfect target for our work. The special issue invites research articles, review articles as well as short communications. According the Reviewers valuable suggestion we revised the hypothesis to” Therefore, the aim of the present study was to test the hypothesis that central venous line placement and tip confirmation can safely be performed via point of care ultra-sound visualization, reducing radiation exposure in pediatric patients undergoing car-diothoracic surgery. Research article may be the appropriate article type to choose.
My only concern is about the use of X-ray to confirm CVC position after the surgery, as authors stated that reducing patient exposure to radiation would be beneficial and achievable using ultrasound. So, despite using ultrasound, do patients will need to undergo X-ray to confirm CVC position or it was only performed in this study to validate their results? In case ultrasound is finally used as a standard for CVC placement, would it be also needed to performed X-ray likewise?
The reviewer addresses the key question of our investigation. The present study was designed to test the hypothesis that tip confirmation can be performed by point-of-care ultrasound diagnostic. Currently, post interventional X-ray is recommendet for the confirmation of tip positioning and exclusion of pneumothorax. We proof that tip positioning can be performed via point-of-care ultrasound and others have that a pneumothorax can also be diagnosed via point-of-care ultrasound (PMID 23103845). Therefore, point-of-care ultrasound may have the potential replace X-ray diagnostic and become the standard diagnostic after CVC insertion. However, after cardiac surgery, there are various indications to perform X-ray diagnostic. Nevertheless, frequently patients require CVC reinsertion during ICU treatment and in the situation when a Xray diagnostic is performed only for confirmation of tip positioning and exclusion of pneumothorax after CVC insertion point-of-care ultrasound diagnostic may have the potential reducing radiation. To highlight this, we have added the following paragraph to our discussion: Furthermore, chest x-ray diagnostics are performed after cardiac surgery not only evaluating the lungs and confirm CVC tip positioning, but also evaluating the heart and chest wall. However, during ICU therapy most often the CVC need to be replaced and our protocol may have the potential reducing radiation in situations when radiography diagnostics and only performed for tip CVC positioning verification and pneumothorax exclusion.
Minor issues:
- Abstract: line 14: remove “potentially” before cardiogenic.
Has been removed
- Abstract: line 16: “can be performed fast and safe by…”
Has been revised
- Abstract: line 17: please define “IRB”.
Another reviewer requested to omit IRB from the abstract
- Abstract: line 17: “inform consent obtainment”
Also omitted upon other reviewers request
- Abstract: line 18: “four probe positions”
Has been revised
- Abstract: line 19: “cardiothoracic surgery”
Has been revised
- Abstract: line 23: following initial placement, and to reduce radiation exposure”
Has been revised
- Abbreviate central venous catheter as CVC all through the manuscript: see lines 22, 81, 197, 208, 225.
Abbreviation has been used throughout the manuscript
- Introduction: line 35: standardized.
Has been revised
- Results: line 182: “the CVC tip or the right pulmonary artery could…”
We are very sorry, but we are not quite sure what exactly the reviewer would like to be changed.
- Discussion: lines 197 and 207. are contradictory: at the beginning authors stated that ultrasound-guided puncture of central vessels is an accepted standard procedure for the insertion of CVC in adults as well as in pediatric patients (7-9). Then, they stated that ultrasound-guided positioning of the CVC tip has been described for adults, but it is not a standard procedure in children (5, 12). I guess it is not accepted as a standard procedure in children, so authors must correct it and clarify.
We would like to thank the Reviewers bringing this to our attention. We have revised the discussion to “Ultrasound-guided positioning of the CVC tip has been described in adults, but studies in children have not been performed so far.” clarifying our statement.
- Discussion: line 210: Please define “PW”.
We have erased this abbreviation as it is no standard and written “pulsed wave doppler”
- Figures 6a, 6b: Please detailed to which aspect percentages refe
We have added the following legend to figure 6 “Figure 6: Visualization of CVC tip and anatomical structures in all Patients (a) and Patients with previous surgery (% all patients in respective group).”
We would like to thank the reviewer for the time he devoted to our submission and his valuable suggestions. We feel that addressing his comments substantially improved the quality of our manuscript.

Reviewer 2 Report
Thank you for giving me an opportunity to review this article.
The authors presented a valuable tool that can be used in clinical practice. However, detailed pre-specifications of the study design and sample size determination are lacking and the small sample size itself can be a limitation of this study. Also, the following issues should be reconsidered.
- The main theme of point-of-care is “fast assessment at the site”. Therefore, focusing on this theme in the background of the abstract would be better rather than describing the chronic consequences of radiation exposure.
- The statement for IRB approval can be omitted in the abstract
- 92,6% --> 92.6%
- A more detailed description regarding methodology rather than background would be better for the abstract. Especially which combination of the views or criteria were used and which outcome was primarily assessed to evaluate the accuracy of the protocol.
- Can a reference regarding ‘RPA as a landmark for SVC-RA junction’ be provided?
- To identify the target population and evaluate the applicability or generalizability of the protocol, a detailed description regarding inclusion and exclusion criteria is needed.
- Line 80 to 81 needs to be revised. The meaning is unclear. The wording ‘only children~’ and subsequent ‘and children~’ seems contradictory.
- Again, to evaluate the applicability or generalizability of the protocol, the profile of the examiners in the study (e.g. an anesthesiologist who has ~years of experience in pediatric sonography) needs to be stated.
- Sample size determination should be stated. Or at least, justification is needed.
- ‘PE’ in fig.3 needs clarification.
- Red underlines are marked in the figures. It should be deleted.
- Figure 4. J wire is not clearly seen in the figure. Does the wire come from Lt IJV? To my knowledge, the wire inserted from Rt IJV should be seen in innominate V and run into SVC.
Also, the terminology VCS used in the figure is quite confusing (VCS or SVC?). To clarify this, a detailed description of the orientation in the figure (e.g. caudal, cephalad, right or left innom V, right or left IJV) is recommended.
- The caliper indicated in Figure 5 needs an explanation
- Table 2 is confusing. What does “mean” stand for? Is it sonographic grading? If so, then why the results were stratified into two categories? It is obvious that insufficiently identifiable cases had higher grades and sufficiently identifiable cases had lower grades.
- Line 182, “could or the or the” needs correction
- Table 2 and Fig 6 need refinement. As 100 – insufficient ratio = sufficient ratio, describing both ratios seems redundant. Also, repeated subheadings (i.e. suprasternal notch svc, suprasternal notch CVC tip) should better be simplified. Also, the results could be stratified by RPA/tip/SVC instead of subcostal/suprasternal/parasternal. Either way, the description of the results needs to be simple and intuitive.
- Was there any adjustment of tip position when too deep or shallow positioning of the tip was detected during the sono exam? If not, is there any added value of point-of-care exam than radiograph?
- In figure 7. A footnote for description and interpretation for the doppler exam is needed.
Author Response
The authors presented a valuable tool that can be used in clinical practice. However, detailed pre-specifications of the study design and sample size determination are lacking and the small sample size itself can be a limitation of this study. Also, the following issues should be reconsidered.
We would like to thank the reviewer for his pleasant evaluation and the time he invested into our submission. We will address his valuable suggestions in a point-by-point response.
- The main theme of point-of-care is “fast assessment at the site”. Therefore, focusing on this theme in the background of the abstract would be better rather than describing the chronic consequences of radiation exposure.
We would like to thank the reviewer for his valuable suggestion. Accordingly, we have included both, the “fast assessment at the site” and potential reduction of radiation and revised the abstract to: “Central venous catheters (CVC) are commonly required for pediatric congenital cardiac surgeries. Current standard for verification of CVC positioning following perioperative insertion is postsurgical radiography. However, incorrect positioning may induce serious complications, including pleural and pericardial effusion, arrhythmias, valvular damage, or incorrect drug re-lease and point of care diagnostic may prevent these serious consequences. Furthermore, pediatric patients with congenital heart disease receive various radiological procedures.”
- The statement for IRB approval can be omitted in the abstract
Has been omitted accordingly
- 92,6% --> 92.6%
Has been changed
- A more detailed description regarding methodology rather than background would be better for the abstract. Especially which combination of the views or criteria were used and which outcome was primarily assessed to evaluate the accuracy of the protocol.
We have included the evaluated probe positions and outcome into the abstract. “We evaluated a point-of-care ultrasound protocol, consistent of the combination of parasternal craniocaudal, parasternal transversal, suprasternal notch, and subcostal probe positions, to verify tip positioning in any of the evaluated views at initial CVC placement in pediatric patients undergoing cardiothoracic surgery for congenital heart disease.”
- Can a reference regarding ‘RPA as a landmark for SVC-RA junction’ be provided?
PMID: 17804772 reference has been included into the manuscript
- To identify the target population and evaluate the applicability or generalizability of the protocol, a detailed description regarding inclusion and exclusion criteria is needed.
The following paragraph “19 children 3 days to 4 years of age and 2.55 to 16 kg body weight that underwent pediatric cardiac surgery for congenital heart disease were included.” has been added specifying included patients.
- Line 80 to 81 needs to be revised. The meaning is unclear. The wording ‘only children~’ and subsequent ‘and children~’ seems contradictory.
Thank you very much. We have revised the text to “Children requiring a CVC for cardiac surgery were included...”
- Again, to evaluate the applicability or generalizability of the protocol, the profile of the examiners in the study (e.g. an anesthesiologist who has ~years of experience in pediatric sonography) needs to be stated.
We have included the following statement “…evaluations were performed by anesthesiologists highly experienced in both, pediatric cardiac anesthesia and vascular ultrasound.” to the method section
- Sample size determination should be stated. Or at least, justification is needed.
Other studies evaluating the feasibility of an ultrasound-guided right subclavian vein (RScV) CVC tip positioning included 20 patients. Therefore we included a similar amount of patients into our study. This information has been included into the method section.
- ‘PE’ in fig.3 needs clarification.
Figure 3 is showing a pericardial effusion (PE). This information has been added to the figure legend
- Red underlines are marked in the figures. It should be deleted.
Have been deleted accordingly
- Figure 4. J wire is not clearly seen in the figure. Does the wire come from Lt IJV? To my knowledge, the wire inserted from Rt IJV should be seen in innominate V and run into SVC.
Legend has been revised according to reviewers suggestion
Also, the terminology VCS used in the figure is quite confusing (VCS or SVC?). To clarify this, a detailed description of the orientation in the figure (e.g. caudal, cephalad, right or left innom V, right or left IJV) is recommended.
We would like to thank the reviewer for catching this. We have revised the nomenclature and amended VCS to SVC throughout the manuscript
- The caliper indicated in Figure 5 needs an explanation
Explanation has been included into the figure legend
- Table 2 is confusing. What does “mean” stand for? Is it sonographic grading? If so, then why the results were stratified into two categories? It is obvious that insufficiently identifiable cases had higher grades and sufficiently identifiable cases had lower grades.
As indicated in the methods the sonographer had to grade the quality of the structures. The mean is the statistical grading quality according. The Reviewer is correct, a higher grade indicates a less sufficient quality.
- Line 182, “could or the or the” needs correction
Again, thank you very much for catching this. We have revised the text to “. Thus, in none of the nine children with previous sternotomy neither the CVC tip, nor the right pulmonary artery could be reliably visualized in the parasternal views”
- Table 2 and Fig 6 need refinement. As 100 – insufficient ratio = sufficient ratio, describing both ratios seems redundant. Also, repeated subheadings (i.e. suprasternal notch svc, suprasternal notch CVC tip) should better be simplified. Also, the results could be stratified by RPA/tip/SVC instead of subcostal/suprasternal/parasternal. Either way, the description of the results needs to be simple and intuitive.
We absolutely agree with the reviewer that the results need to be intuitive and changed the table accordingly. However, to be able to verify the tip position the examiner needs to assess all views and different views have different rates of success. In our point of view this information should be given to the reader. Therefore, we kindly request to keep the entire information in the table and hope that the Reviewer agrees with our restruction of the table.
- Was there any adjustment of tip position when too deep or shallow positioning of the tip was detected during the sono exam? If not, is there any added value of point-of-care exam than radiograph?
Yes! As indicated in the method section the primary depth of insertion of the CVC was calculated based on the child's height using the following formula: primarily insertion depth (cm)= body length (cm)/ 10. After distinct visualization of the CVC tip in the SVC, the catheter was corrected for position.
- In figure 7. A footnote for description and interpretation for the doppler exam is needed.
We have included the following “Figure 7: Subcostal view: PW Doppler signal from the superior vena cava with systolic (S), diastolic (D), and atrial (A) waves.” and highlighted the corresponding points in the figure.

Round 2
Reviewer 2 Report
- The nomination in Figure 4 still needs further revision. “Left” and “right” better be capitalized as “Lt” and “Rt”.
- In my point of view, Table 2 and Fig 6 still need refinement. In cases with binary outcome (for example, success or failure), if the success rate is 20%, then the failure rate should be 80%, obviously. Therefore, there is no extra information gained from presenting both rates over presenting either of the rates. It rather decreases readability. Is there any justification for presenting both rates simultaneously?
- What were the criteria used for the confirmation of correct positioning of the catheter in postoperative chest radiograph? It should be specified in the manuscript.
- Wasn't there any discrepancy between sonographic and radiographic determinations?
Author Response
- The nomination in Figure 4 still needs further revision. “Left” and “right” better be capitalized as “Lt” and “Rt”.
Has been revised according to Reviewer’s suggestion
- In my point of view, Table 2 and Fig 6 still need refinement. In cases with binary outcome (for example, success or failure), if the success rate is 20%, then the failure rate should be 80%, obviously. Therefore, there is no extra information gained from presenting both rates over presenting either of the rates. It rather decreases readability. Is there any justification for presenting both rates simultaneously?
We absolutely agree with this suggestion, omitted the non-success rates and revised the table accordingly
- What were the criteria used for the confirmation of correct positioning of the catheter in postoperative chest radiograph? It should be specified in the manuscript.
The following information „X-ray verification of the CVC tip at the junction of the SVC and the RA” has been added to the manuscript specifying the radiographic evaluation
- Wasn't there any discrepancy between sonographic and radiographic determinations?
No, this is the point. If the CVC tip could be visualized the correct position was achieved by point-of-care ultrasound
Again, we would like to thank the reviewer for his time and suggestions. We hope that we were able to address his concerns.
